# Control of a Micro-Electro-Mechanical System Fast Steering Mirror with an Input Shaping Algorithm

**DOI:** 10.3390/mi15101215

**Published:** 2024-09-29

**Authors:** Jiapeng Hou, Haoxiang Li, Lei Qian, Huijun Yu, Wenjiang Shen

**Affiliations:** 1School of Nano-Tech and Nano-Bionics, University of Science and Technology of China, Hefei 230026, China; jphou@mail.ustc.edu.cn (J.H.);; 2Suzhou Institute of Nano-Tech and Nano-Bionics, Chinese Academy of Sciences, Suzhou 215123, China

**Keywords:** micro-electro-mechanical system (MEMS), fast steering mirror (FSM), overshoot suppression, input shaping

## Abstract

Fast steering mirrors (FSMs) designed by the micro-electro-mechanical system (MEMS) technology are significantly smaller in volume and mass, offering distinct advantages. To improve their performance in the open-loop control mode, this study introduces a control algorithm and evaluates its performance on an electromagnetic-driven MEMS-FSM. The algorithm employs a method to shape the input signal by fitting the system’s transfer function and modifying the step response. This shaped signal is then applied to the system to minimize overshoot, reduce settling time, and improve working bandwidth, thereby enabling faster angular adjustments and improving the stability of the FSM. The experimental results demonstrate an 85.65% reduction in overshoot and a decrease in settling time from 84 ms to 0.4 ms. Consequently, the working bandwidth of the FSM system increases to 2500 Hz, demonstrating the effectiveness of the algorithm in enhancing MEMS-FSM’s performance.

## 1. Introduction

The fast steering mirror (FSM) is a critical optical device for achieving rapid laser beam control, extensively used in electro-optic systems such as inter-orbit satellite laser communication, image stabilization, and astronomical telescopes [1,2]. In inter-orbit satellite laser communication, the FSM captures the target satellite and adjusts the laser beam’s deflection angle in real time. By this means, FSM can point and track the satellite’s laser receiver. However, factors such as the vibration, instability, and positioning error of the satellite platform demand high precision and stability from the FSM system [3]. There are currently two main drive methods: voice coil motor drive and Lead Zirconate Titanate (PZT) stacking drive. The disadvantage of the voice coil motor driving method is that the motion accuracy is limited. Since the voice coil motor relies on electromagnetic induction to generate movement, the precision is constrained by the motor’s control accuracy and mechanical design. This limitation is particularly evident in applications requiring ultra-high precision positioning, where it may fail to meet the requirements [4]. Additionally, the FSM driven by PZT stacking faces issues of response non-linearity due to hysteresis, and temperature variations can also affect the performance of the piezoelectric ceramics [5]. Furthermore, these methods face challenges in miniaturization and integration due to the limited size of the drivers. In contrast, FSMs produced by the micro-elector-mechanical system (MEMS) technology are significantly smaller in volume and mass, offering advantages in inter-orbit satellite communication by better meeting application requirements [6,7]. For improving the performance in the open-loop control mode, this study introduces a control algorithm and evaluates its performance on an electromagnetic-driven MEMS-FSM.

To minimize vibrations, instability, and positioning errors during orbit, the FSM operates in a quasi-static mode [8]. This mode has frequencies typically ranging between tens and hundreds of Hz. The working bandwidth is an important parameter for the quasi-static mode. It is the reciprocal of the system’s settling time and reflects the FSM’s ability to adjust laser pointing rapidly. To enhance FSM performance, appropriate control algorithms must be developed based on the FSM system’s characteristics [9,10]. In past research, Kerboua M. et al. explored passive piezoelectric vibration shunt control for cantilever beams to suppress system overshoot, though MEMS-FSM’s small size complicates integration, increases manufacturing complexity, and raises costs [11]. Additionally, Imboden M. et al. demonstrated overshoot suppression and rapid stabilization in MEMS mirrors using a two-step algorithm [12]. Corey P. utilized overdrive algorithms to shorten the settling time of the MEMS-FSM and enhance its quasi-static performance [13]. Yu Z. et al. achieved significant results in suppressing overshoot in piezoelectric MEMS-FSMs using a similar approach [14]. Although the overdrive algorithm has been demonstrated to be effective in suppressing overshoot in MEMS-FSM, the acquisition of the overdrive signal remains challenging.

This study introduces a novel input signal-shaping algorithm, allowing for an arbitrary modification of the system’s response, while simplifying the acquisition of control signals, thereby enhancing control flexibility. The algorithm was validated on an electromagnetic-driven MEMS-FSM. The FSM’s response was measured by a position-sensitive detector (PSD) sensor and the transfer function fitted via the least squares method. The input signal-shaping filter was implemented, and the filtered control signal was generated by a programmable signal generator. The FSM system’s overshoot was reduced by 85.65%, and the settling time was reduced from 84 ms to 0.4 ms, resulting in an increased operating bandwidth of 2500 Hz. By minimizing the FSM overshoot, the algorithm improves quasi-static performance, achieving fast angle switching and stability.

This paper is organized as follows. Section 2 introduces the FSM’s structure, operating principle, and dynamic analysis; Section 3 discusses the algorithm’s design; Section 4 presents the algorithm’s implementation and analyzes the input signal shaping and response results. Finally, Section 5 concludes this study.

## 2. FSM System Analysis

### 2.1. Structure of MEMS-FSM

The FSM comprises a mirror and multiple driving coils. The structure of the electromagnetic-driven MEMS-FSM is illustrated in Figure 1a. All are integrated onto a silicon substrate-based driver through MEMS processing techniques as shown in Figure 1b. The driver is supported by four serpentine springs connected to the surrounding frame. A magnetic field, supplied by permanent magnets within the package, interacts with the coils to facilitate mirror movement. The mirror is mounted on the driver via a pillar, which facilitates the selection of different mirror sizes and coatings based on specific application requirements. Figure 1c shows the MEMS-FSM with a 7.5 mm square mirror.

Compared with piezoelectric drives, electromagnetic drives offer better linearity in MEMS-FSM. Additionally, the electromagnetic drive method excels over electrostatic drives in terms of impact resistance, meeting the stringent requirements for aerospace-grade impact resistance [15,16]. When an electric current flows through the electromagnetic-driven MEMS-FSM, the coil within the permanent magnet’s magnetic field experiences a vertical electromagnetic force, causing the mirror to deflect. Two orthogonally arranged coils are controlled independently, enabling a two-dimensional mirror deflection.

For small broadband inter-orbit satellites, where space is limited, a large mass and volume can lead to increased launch costs [17]. The voice coil motor FSM (V-931) and PZT FSM (S-330) produced by the German company PI weigh over 500 g and have thicknesses of 26 mm and 42 mm, respectively [18,19]. In comparison, FSMs manufactured using the MEMS technology are significantly smaller in both volume and mass, as shown in Table 1. This compact size offers substantial advantages for inter-orbit satellite communication, better fulfilling the development needs of the field.

### 2.2. Basic Principle

The FSM operates as a second-order spring-damping system. The elastic torsion axis constrains the mirror, allowing it to undergo torsional motion during operation. This motion is analogous to the compression and extension of a spring. Additionally, the mirror is influenced by both structural damping and air damping. As shown in Figure 2, the dynamic behavior of the FSM can be represented by a simplified model of a second-order spring-damping system. In this model, the spring has an elastic coefficient *k*, and the damper has a damping coefficient *b*, both of which are connected to a mass *m*.

### 2.3. Dynamic Analysis

As analyzed above, the dynamic equation of the second-order spring damping system is expressed as Equation (Equation 1): (1)$$m\ddot{x}+b\dot{x}+kx=F.$$

Its open-loop transfer function is expressed with Equation (Equation 2): (2)$$\phi \left(s\right)=\frac{\omega_{n}^{2}}{s^{2}+2\zeta \omega_{n}s+\omega_{n}^{2}}.$$

In Equation (Equation 2), $\omega_{n}$ is the undamped natural frequency, and ζ is the damping ratio—a dimensionless parameter that characterizes the decay rate of the system’s vibrations. When ζ<1, the system is considered underdamped. It is widely recognized that a step signal input represents a challenging condition for system performance. If the system shows satisfactory dynamic behavior under step signal excitation, it is likely to achieve an optimal dynamic response under other inputs as well. The typical step response of a second-order underdamped system is illustrated in Figure 3.
(3)$$\sigma \%=\frac{h\left(t_{p}\right)-h\left(\infty \right)}{h(\infty )}\times 100\%.$$

In Equation (Equation 3), σ denotes the overshoot and is defined as the percentage by which the step response peak $h\left(t_{p}\right)$ exceeds the final value $h\left(\infty \right)$. The settling time named $t_{s}$ is defined as the shortest time required for the step response to enter and remain within a 5% (or 3%) error band of the final value $h\left(\infty \right)$. This parameter reflects the system’s ability to quickly achieve a stable state.

## 3. Algorithm Design

This study treats the FSM system as a “black box”, focusing solely on the relationship between the input signal and the MEMS-FSM’s response. The deflection angle of the FSM is converted into the displacement of the spot on the PSD sensor (e-otron, Shanghai, China). A comprehensive description of the system’s behavior is obtained by fitting the transfer function of the FSM system. In an open-loop control configuration, the characteristics of the FSM system remain unchanged during operation. This situation means the transfer function remains stable. Based on the above, an open-loop control algorithm for the FSM system was developed to reduce overshoot and settling time. This algorithm can facilitate rapid angle switching and enhance the stability of the FSM system.

The algorithm’s workflow is illustrated in Figure 4. Initially, the FSM system’s step response is recorded via the PSD sensor, and the transfer function *A* is estimated using the least squares method. Next, the step response curve of the FSM system is modified. The purpose of this modification is to compress the response points near the final value, thereby reducing oscillations and minimizing overshoot, leading to an ideal step response for the FSM system. Subsequently, the transfer function *B* is refitted by the least squares method. Then, the transfer function *C* = *B*/*A* is evaluated. It means that *C* × *A* is the transfer function *B*, which gives the ideal response. The response overshoot is minimized, and the settling time is reduced by shaping the input signal through a filter *C*. This strategy can also improve the working bandwidth and fast switching angle.

By importing the input signal and response of the FSM system and specifying the order of the transfer function, a transfer function model of the FSM system can be obtained using the least squares method [20,21]. The core principle of this method is to minimize the sum of squared errors (SSE), thereby reducing the discrepancy between the predicted values and the actual observations of the model [22,23]. For a given data point $\left(x_{i},y_{i}\right)$, the predicted value from the fitting model is $\hat{y_{i}}$, and the error is defined as shown in Equation (Equation 4)
(4)$$e=y_{i}-\hat{y_{i}}.$$

For second-order systems with two poles and at most one zero, the transfer function typically takes the form shown in Equation (Equation 5).
(5)$$H\left(s\right)=\frac{b_{0}s+b_{1}}{s^{2}+a_{1}s+a_{2}},$$

In Equation (Equation 5), $b_{0}$, $b_{1}$, $a_{1}$, and $a_{2}$ are parameters that need to be estimated, represented by the vector θ. The least squares method aims to minimize the sum of squares of all errors, which can be expressed as an optimization problem in Equation (Equation 6).
(6)$$SSE_{\min}=\sum_{i=1}^{n}e_{i}^{2}\left(\theta \right)=\sum_{i=1}^{n}{\left(y_{i}-\hat{y_{i}}\left(\theta \right)\right)}^{2}.$$

The algorithm adjusts the vector θ using gradient descent. Initially, the parameter vector $\theta^{\left(0\right)}$ is initialized, and the gradient $\nabla E(\theta^{\left(k\right)})$ of the error function with respect to the parameters is calculated. The parameters are then updated along the negative gradient direction with a learning rate α. The update formula is given in Equation (Equation 7).
(7)$$\theta^{\left(k+1\right)}=\theta^{\left(k\right)}-\alpha \nabla E\left(\theta^{\left(k\right)}\right).$$

The iteration process continues until the parameter updates become negligible or the maximum number of iterations is reached. The final result is the fitted transfer function [24].

As shown in Figure 3, the step response of an underdamped second-order system reaches its peak within the first half-period, followed by oscillations around the final value and gradual decay. To modify the step response, the process begins by identifying the first point where the response exceeds the final value. The difference between subsequent responses and the final value is multiplied by a desired attenuation coefficient and then added to the final value. This approach compresses the response curve near the final value, reducing overshoot and settling time. The specific implementation is shown in Equation (Equation 8). In this equation, $t_{1}$ is the time when the response first exceeds the final value, *d* is the attenuation coefficient, and $h\left(\infty \right)$ is the final value. The step responses after modification are illustrated in Figure 5.
(8)$$h\left(t_{1,2,\ldots ,n}\right)=\left(h\left(t_{1,2,\ldots ,n}\right)-h\left(\infty \right)\right)\times d+h\left(\infty \right).$$

## 4. Experiment and Result

### 4.1. Experimental Setup

The experimental setup for this study is illustrated in Figure 6, utilizing a 650 nm wavelength laser with a power output of 5 mW. By adjusting the relative position of the FSM and the laser, the reflected beam spot was directed onto a PSD sensor. The movement trajectory of the spot on the PSD sensor, as observed through a host computer, represents the FSM’s response to the input signal. A standard step signal with a normalized amplitude of 1 was generated using a programmable signal generator. Since the tested FSM lacks universal joint structures and exhibits identical behavior in both orthogonal directions, validation was performed along a single axis. The system response recorded by the PSD sensor is shown in Figure 7a, where the PSD sensor’s sampling frequency was set at 2.5 kHz, corresponding to a time interval of 0.4 ms between consecutive data points. The response parameters of the MEMS-FSM system are summarized in Table 2. The FSM system exhibits a large overshoot and long settling time. To achieve rapid, faster angular adjustments and improved stability, the FSM system’s overshoot must be minimized.

### 4.2. Parameter Identification and Input Shaping

The transfer function of the FSM system is fitted using the least squares method. A standard step signal is applied to the fitted transfer function, and the response is compared with the original response curve, as shown in Figure 7b. The response curve is modified using Equation (Equation 5) to obtain the ideal response curve, as illustrated in Figure 7c. The ideal response transfer function is then fitted using the least squares method, and a standard step signal is applied to this fitted ideal transfer function. The response is compared with the original ideal response curve, as shown in Figure 7d.

Following the algorithm’s workflow shown in Figure 4, the transfer functions *A* for the original FSM system and *B* for the ideal FSM system were derived. Based on the relationship between time-domain and frequency-domain signal transformations, the transfer function *C* of the input signal filter was obtained by convolving the inverse of transfer function *B* with transfer function *A*. The parameters for the original FSM system transfer function *A*, the ideal FSM system transfer function *B*, and the input signal filter transfer function *C* are detailed in Table 3. If the response obtained by the filter is applied to the FSM system, the output of the FSM system will match the response of an ideal FSM system when a step signal is directly input. This phenomenon verifies that the algorithm proposed in this article can effectively reduce the overshoot. The Final Prediction Error (FPE) and Mean Square Error (MSE) quantify the fitting accuracy of the transfer function, with smaller values indicating a closer match to the system’s actual transfer function. As shown in Table 3, the fitted transfer function demonstrates high confidence.

In practical applications, electromagnetic MEMS-FSMs operate in a quasi-static mode when driven by step signals of varying amplitudes. The input step signal is provided by a programmable signal generator, causing the FSM to deflect at different angles to control the laser beam’s deflection direction entering the mirror. As shown in Figure 8, the input step signal voltage ranges from −1 V to 1 V, and the displacement of the FSM’s reflected beam is linearly correlated with the amplitude of the step signal [25]. After determining the input signal-shaping filter, which uses *C* as the transfer function, different amplitudes of step signals are input into the filter to yield the ideal input signal, as shown in Figure 9. If this ideal input signal is fed into the system, it will produce the ideal response.

### 4.3. Experimental Results

The ideal input signal can be applied to the FSM system via a programmable signal generator, and the FSM system’s response is captured using a PSD sensor. Figure 10 illustrates the FSM system’s response to the shaped step signals. The parameters of the FSM system’s response to a standard step signal in Table 4. As shown in Table 4, with the addition of the input signal-shaping filter, the FSM system was significantly improved in terms of overshoot reduction and settling time decrease. The overshoot of the FSM system decreased by 85.65%, and the settling time was reduced from 84 ms to 0.4 ms. Furthermore, as the settling time decreased, the bandwidth of the FSM system increased to 2500 Hz, enabling fast angle switching and stability. Since the displacement of the FSM’s reflected spot is linearly related to the amplitude of the step signal, the input signal-shaping filter proved effective across various step signal amplitudes, confirming its robustness in practical applications.

### 4.4. Result Discussion

Although the algorithm attenuates the original system response with a 1% attenuation coefficient, practical testing revealed that the overshoot did not align with the algorithm’s expectations. The algorithm’s performance relies on the accurate transfer function of the FSM system and high precision fitting of the modified response. Several potential sources of error may affect the results during practical testing. Firstly, when acquiring the step response of the FSM system by the PSD sensor, the sampling rate may be limited. Additionally, the PSD sensor might be susceptible to ambient light noise. This may lead to discrepancies between the captured response and the actual step response. Secondly, when applying the least squares method to fit the transfer function, neither the Final Prediction Error (FPE) nor the Mean Square Error (MSE) is zero. This situation implies a difference between the fitted transfer function and the actual one. Moreover, when outputting waveforms through a programmable signal generator, the number of output waveform points might be lost due to performance limitations. This may lead to discrepancies between the algorithm’s output signal and the actual input signal applied to the FSM. To address these issues, PSD sensors with higher sensitivity, enhanced noise suppression, and higher sampling rates should be used to acquire the response of the system step. Additionally, the transfer function of the FSM system can be more accurately fitted by employing an improved least squares method and incorporating a neural network algorithm. Finally, using a programmable signal generator with a higher resolution will further minimize errors and enhance the algorithm’s effectiveness.

## 5. Conclusions

This paper introduces a control algorithm based on input signal shaping and validates its application through a custom-designed electromagnetic-driven MEMS-FSM.

Initially, the paper outlines the structure and fundamental principles of the MEMS-FSM. Compared to FSMs driven by voice coil motors or PZT stacks, MEMS-FSMs offer a higher integration, reduced size, and lower mass. These features make them particularly advantageous for inter-satellite communication, meeting the evolving demands of various applications.

To enhance the quasi-static performance of MEMS-FSMs, a method was developed to connect the input signal with the ideal system response by fitting the system’s transfer function and modifying its response. This led to the derivation of a calculation method for the transfer function of the input signal-shaping filter. By using the filter to shape the input signal, the algorithm effectively minimizes system overshoot, reduces settling time, and improves working bandwidth, resulting in the fast angle switching and improved stability of the FSM system. After incorporating the input signal-shaping filter, the FSM system’s overshoot was reduced by 85.65%, the settling time was reduced from 84 ms to 0.4 ms, and the working bandwidth increased to 2500 Hz, confirming the algorithm’s effectiveness. The algorithm allows for the input signal to be shaped to achieve an ideal system response, providing flexible system control.

Importantly, this algorithm is not limited to MEMS-FSM systems. Treating the system as a “black box” simplifies its application, making the algorithm applicable to other systems. Future work will focus on further optimizing the MEMS-FSM response, such as reducing rise time. Additionally, plans include integrating piezoresistive sensors for feedback signal detection within the MEMS-FSM and implementing this algorithm as an input signal filter within the system.

## Figures and Tables

**Figure 1 micromachines-15-01215-f001:**
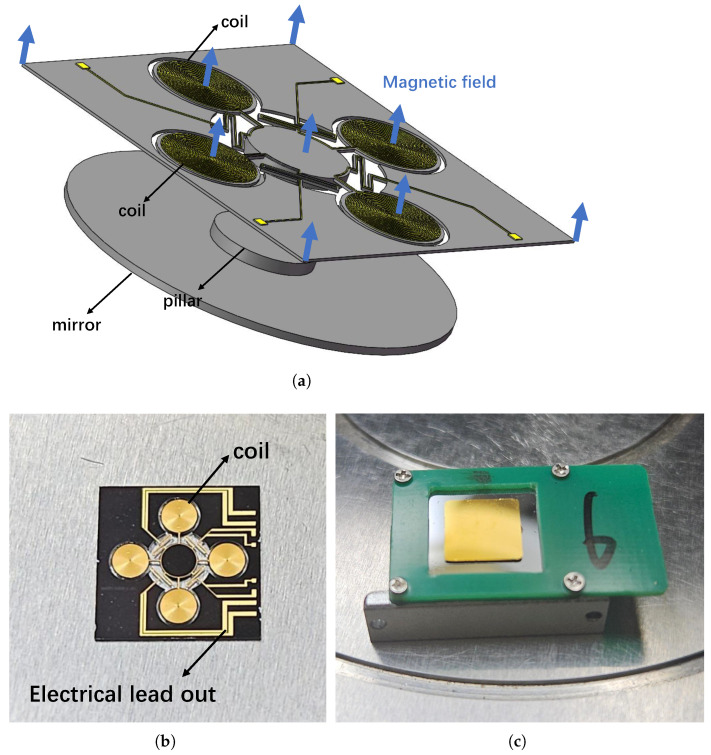
Overview of a MEMS FSM. (**a**) Structure of the MEMS-FSM; (**b**) The driver part of the MEMS-FSM; (**c**) MEMS-FSM with a 7.5 mm square mirror.

**Figure 2 micromachines-15-01215-f002:**
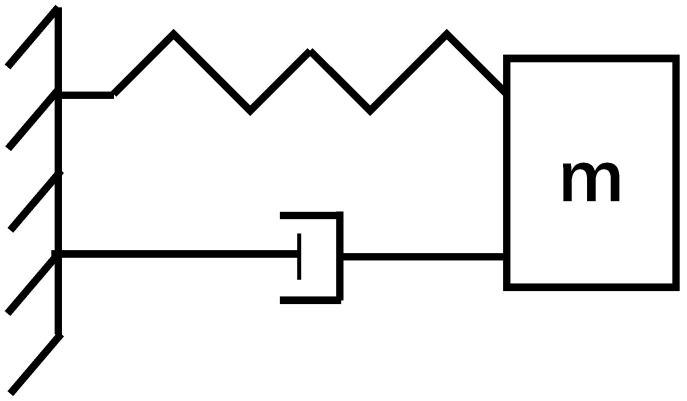
Simplified model of a second-order spring damping system.

**Figure 3 micromachines-15-01215-f003:**
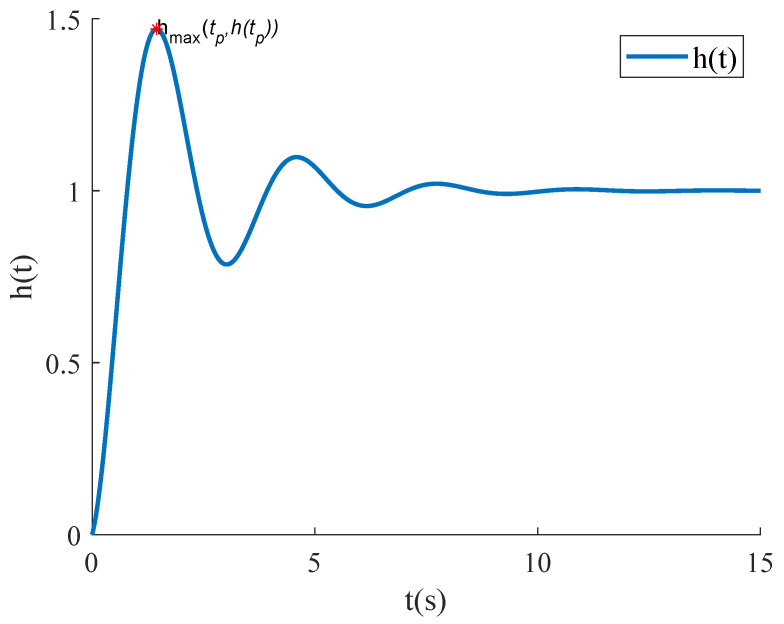
Typical step response of a second-order underdamped system.

**Figure 4 micromachines-15-01215-f004:**
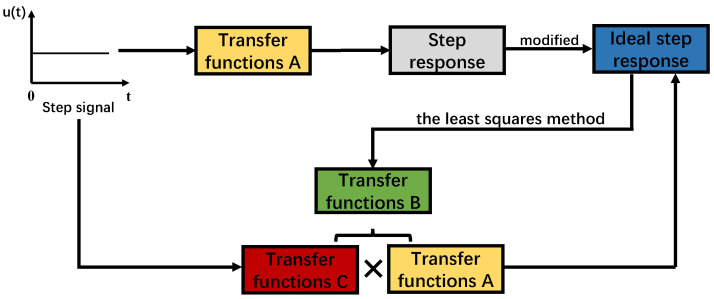
Algorithm’s workflow.

**Figure 5 micromachines-15-01215-f005:**
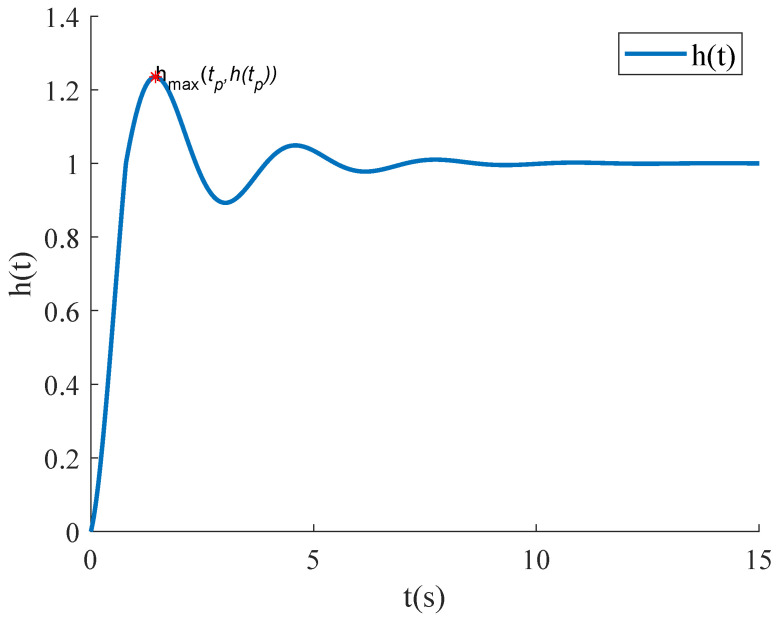
Typical step response after modification.

**Figure 6 micromachines-15-01215-f006:**
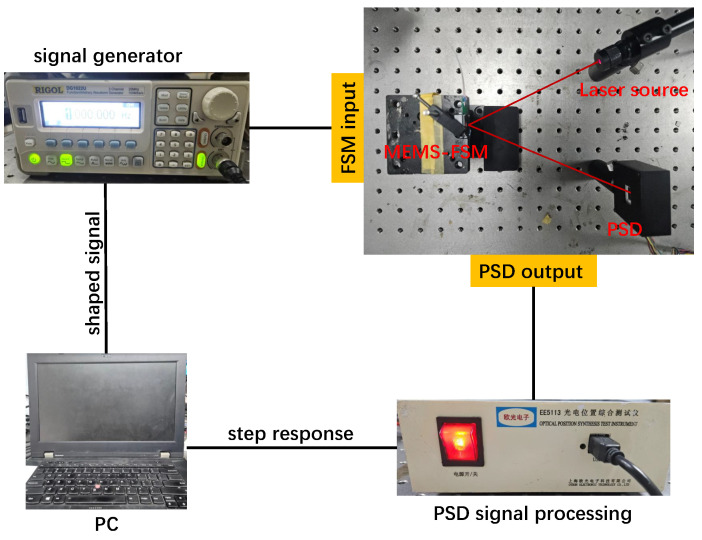
FSM test system.

**Figure 7 micromachines-15-01215-f007:**
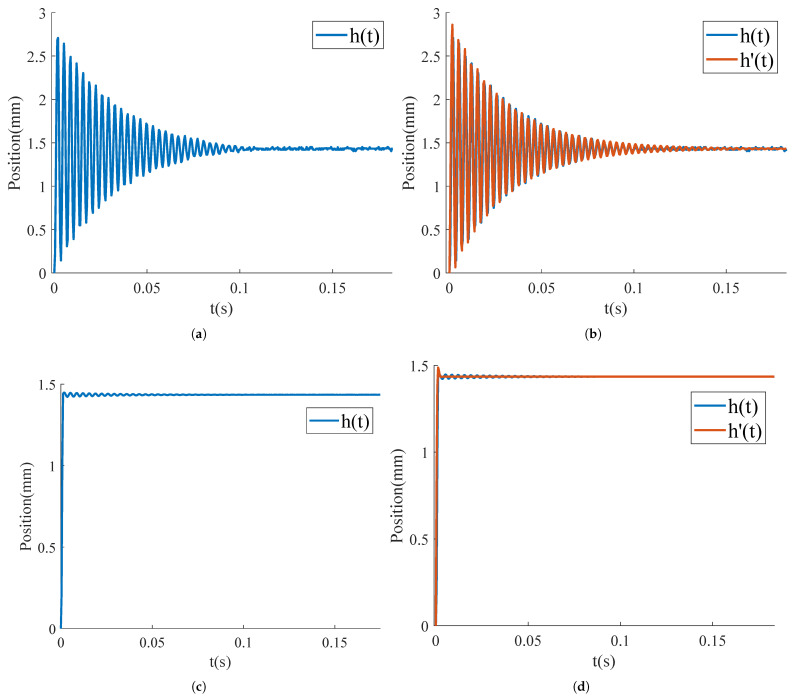
FSM system response and modification. (**a**) FSM system response curves; (**b**) Fitted FSM system response curves; (**c**) Response curve of the modified FSM system; (**d**) Fitting of the response curve of the modified FSM system.

**Figure 8 micromachines-15-01215-f008:**
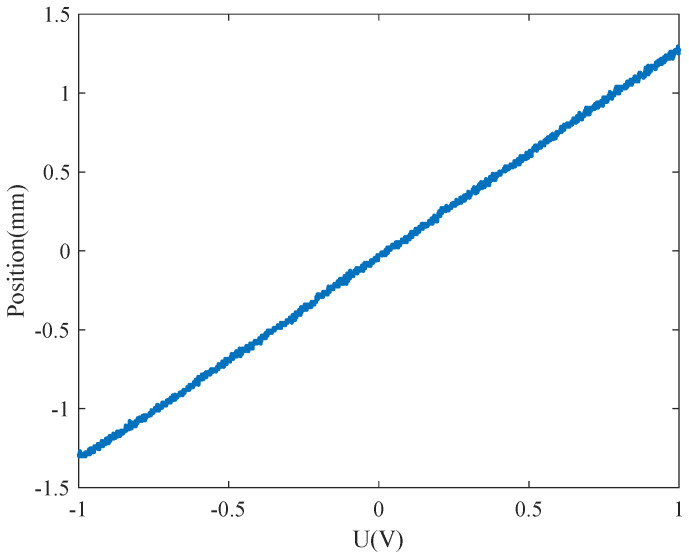
The relationship between the spot reflected by FSM and the input voltage.

**Figure 9 micromachines-15-01215-f009:**
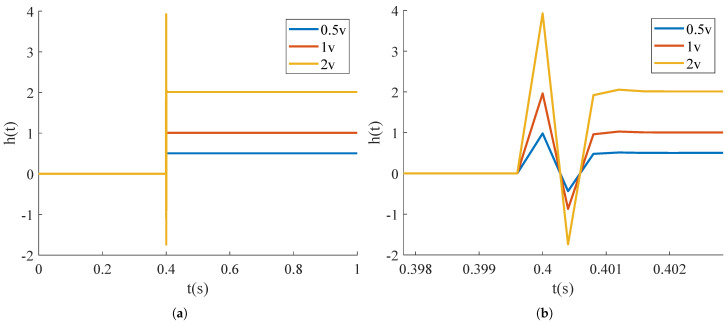
The step signal after being shaped. (**a**) Shaping of step signals with different amplitudes; (**b**) A closer look at the curves.

**Figure 10 micromachines-15-01215-f010:**
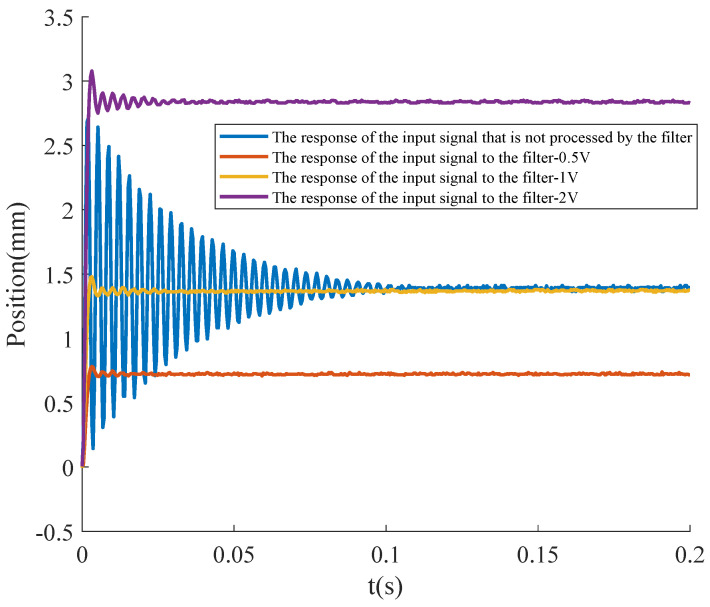
The response of the system.

**Table 1 micromachines-15-01215-t001:** Structural parameters of MEMS-FSM.

Structure	Parameter
mirror diameter	7.5 mm
chip size	7.5 mm × 7.5 mm × 0.3 mm
chip mass	3 g
tilt angle	±17.45 mrad
angular accuracy	1 μrad

**Table 2 micromachines-15-01215-t002:** Response parameters of the FSM system.

Parameter	Value
$h\left(t_{p}\right)$	2.708 mm
$h\left(\infty \right)$	1.435 mm
σ%	88.71%
$t_{s}$	84 ms

**Table 3 micromachines-15-01215-t003:** Parameters for each transfer function.

	Numerator	Denominator	FPE	MSE	Confidence
*A*	$0.7278z^{-1}$	$1-1.465z^{-1}+0.9736z^{-2}$	$1.590\times 10^{-4}$	$1.575\times 10^{-6}$	93.90%
*B*	$0.8916z^{-1}$	$1-0.538z^{-1}+0.1594z^{-2}$	$6.853\times 10^{-6}$	$6.785\times 10^{-6}$	95.72%
*C*	$0.8916z^{3}-1.3063z^{2}+0.8681$	$0.7278z^{3}-0.3916z^{2}+0.116$	/	/	/

**Table 4 micromachines-15-01215-t004:** The parameters of the FSM system’s response.

	$h(t_{p})$	σ%	$t_{s}$
open loop	2.708 mm	88.71%	84 ms
input shaped	1.479 mm	3.06%	0.4 ms

## Data Availability

The original contributions presented in the study are included in the article, further inquiries can be directed to the corresponding authors.

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
