# Peer review of "Control of a Micro-Electro-Mechanical System Fast Steering Mirror with an Input Shaping Algorithm"

_micromachines, 2024, doi:10.3390/mi15101215_

Round 1

Reviewer 1 Report

Comments and Suggestions for Authors

1. The review of related works in the Introduction is insufficient, particularly regarding the advantages and limitations of different drive methods and control algorithms.

2. The study utilizes an open-loop control algorithm, but it would be beneficial to include a discussion on why closed-loop control was not adopted.

3. In Section 2, the description of the fundamental principles appears disconnected from the paper, such as the explanation of the concept of overshoot, seem superfluous and unnecessary.

4. As for Figure 6, the picture appears to have been taken casually. It is recommended to adjust the pose of the camera and apply more formal post-processing, meanwhile, clearly labeling the specific equipment shown in the picture.

5. In Section 4, when evaluating the algorithm performance, it would be helpful to include qualitative comparisons rather than relying solely on graphical results.

Reviewer 2 Report

Comments and Suggestions for Authors

Dear Authors, I think that there are a few issues which are not clear.

I have highlighted them in the attached file.

Please, give the explanation. 
